# Adherence to Food-Based Dietary Guidelines: A Systemic Review of High-Income and Low- and Middle-Income Countries

**DOI:** 10.3390/nu13031038

**Published:** 2021-03-23

**Authors:** Ana Carolina B. Leme, Sophia Hou, Regina Mara Fisberg, Mauro Fisberg, Jess Haines

**Affiliations:** 1Department of Family Relations and Applied Nutrition, University of Guelph, Guelph, ON N1G 2W1, Canada; shou01@uoguelph.ca (S.H.); jhaines@uoguelph.ca (J.H.); 2Center of Excellence in Nutrition and Feeding Difficulties, PENSI Institute, Sabará Children’s Hospital, José Luis Egydio Setúbal Foundation, São Paulo 01228-200, Brazil; mauro.fisberg@pensi.org.br; 3Department of Nutrition, School of Public University of São Paulo, São Paulo 01246-904, Brazil; regina.fisberg@gmail.com

**Keywords:** dietary guidelines, adherence, diet intake, high-income countries, low- and middle-income countries

## Abstract

Research comparing the adherence to food-based dietary guidelines (FBDGs) across countries with different socio-economic status is lacking, which may be a concern for developing nutrition policies. The aim was to report on the adherence to FBDGs in high-income (HIC) and low-and-middle-income countries (LMIC). A systematic review with searches in six databases was performed up to June 2020. English language articles were included if they investigated a population of healthy children and adults (7–65 years), using an observational or experimental design evaluating adherence to national FBDGs. Findings indicate that almost 40% of populations in both HIC and LMIC do not adhere to their national FBDGs. Fruit and vegetables (FV) were most adhered to and the prevalence of adhering FV guidelines was between 7% to 67.3%. HIC have higher consumption of discretionary foods, while results were mixed for LMIC. Grains and dairy were consumed below recommendations in both HIC and LMIC. Consumption of animal proteins (>30%), particularly red meat, exceeded the recommendations. Individuals from HIC and LMIC may be falling short of at least one dietary recommendation from their country’s guidelines. Future health policies, behavioral-change strategies, and dietary guidelines may consider these results in their development.

## 1. Introduction

The obesity epidemic is becoming the greatest public health concern worldwide. Globally, current data suggest that 1307 million adults are overweight and 671 million are obese, and the number of cases in low- and middle-income countries (LMIC) are rapidly reaching those observed in high-income countries (HIC) [1]. Morbidity and mortality related to obesity have been shown to follow a socio-economic gradient, with higher rates of chronic non-communicable diseases (NCD) among those from lower socio-economic positions [2]. Diet, along with other lifestyle behaviors, is an important risk factor for many NCD, and a large number of dietary components have been shown to be socio-economically patterned [3,4,5]. The sharing of cultural ideas is enhanced with the rapid development of the flow of goods, services, and capital, and the broadening of social networks via advanced communication technologies and enhanced transportation systems. Thus, globalization has accelerated diet and lifestyle changes—as seen with the westernization of diet quality [6].

Although first demonstrated in HIC, changes in diet quality have also been found in LMIC [7,8]. For instance, results from multi-center cross-sectional study, assessing the diet quality of individuals (*n* = 9218) for eight Latin American countries, showed that better scores for healthy eating were found in higher socio-economic populations, while scores for unhealthy diet were observed in lower socio-economic populations [9]. Similar results were found in an Australian population-based study (*n* = 11,247) [10]. Promoting the consumption of a different food sources and high-quality diet among populations across countries with different socio-economic status is an essential challenge to overcome [9,10].

A high diet quality consists of a variety of fruit and vegetables, lean meat and alternatives, low-fat dairy, whole grains, and an adequate ratio of fatty acids (i.e., omega 6 and 3 fatty acids), while minimizing the consumption of discretionary foods, such as those rich in added sugars, saturated fat, alcohol, and sodium [11,12,13]. Country-specific food-based dietary guidelines (FBDGs) are crucial for policy reference standards in food and nutrition, health, and agriculture [11,14]. FBDGs provide individuals dietary advice to promote health, prevent diet-related diseases.

More than 100 countries have developed or are currently developing their FBDG [11,15]. To date, reviews on FBDGs have focused on providing a descriptive summary of current global FBDG, evaluating differences and similarities of key elements of a healthy diet [16]. However, there is gap in reviews that evaluate adherence to FBDGs. Given that adherence to FBDG has important implications for the diet and health of individuals and populations, exploring levels of adherence to FBDG is important to inform public health policies, and behavior-change strategies. Furthermore, understanding how adherence may differ among HIC and LMIC is important given the disparities in diet quality across different parts of the world [17].

Identification of trends and dietary inadequacies and inequalities can help inform more targeted policies and behavior-change strategies to improve population health across different socio-economic countries [18]. However, research comparing adherence to FBDGs are lacking, which may be a concern for the development of nutrition policies. The aim of this study was to review the content of all available FBDGs and report on the adherence to the national FBDGs available in both HIC and LMIC, identified according to the Food and Agriculture Organization/World Health Organization (FAO/WHO) list of guidelines [13]. The comparison between HIC and LMIC adherence to national FBDGs will inform potential areas for improvements in future dietary guidelines.

## 2. Materials and Methods

The protocol for this systematic review was registered with PROSPERO (CRD 42020191131) [19] accessible at https://www.crd.york.ac.uk/prospero/, (17 March 2021) and has been reported according to Preferred Reporting Items for Systematic Reviews and Meta-Analysis (PRISMA) guidelines [20].

### 2.1. Identification of the Studies

A systematic search of six electronic databases (i.e., CINAHL, Lilacs/SciElo, ProQuest, PubMed, Scopus, Web of Science) was performed up to June 2020. Search results indexed within each database from the date of inception to the search date were screened by two authors (AL and SH). The following structured search strings were used: Adult OR Young Adult OR Male OR Female AND Dietary Guidelines OR Food Guidelines OR Dietary Recommendations OR Recommended Diet AND Socio-Economic OR Parental Education OR Paternal Education OR Maternal Education OR Income AND Dietary Intake OR Food Consumption OR Feeding Behavior OR Fruit OR Vegetable OR Fat OR Sodium OR Sugar. Relevant truncations and adjacencies were used to enhance results by allowing variations of the search terms. Manual review of the reference lists was conducted to identify studies that may have been missed. Records were downloaded to EndNote X9.2 and duplicates removed. Records were first assessed by title and abstract and then full text. All records were assessed for inclusion based on the defined criteria. Any uncertainties regarding the inclusion of a study were resolved through discussion among A.L. and S.H. or J.H.

### 2.2. Eligibility Criteria

This review was limited to studies published in English. All studies were assessed according to the following inclusion and exclusion criteria summarized according to the PICO (Participants, Intervention, Comparison, and Outcome) framework:

Participants: Studies were eligible if they included free-living children, adolescents, and adults until 60–65 years. The cut-off of 60 y was used in studies in LMIC and 65 y was used for studies in HIC, which was based on the countries’ definition for older adults. National FBDGs generally focus on these populations [11] as individuals outside of this age range typically have special energy and nutrient needs [21,22] Studies that included participants with a pre-existing disease, an organic cause for obesity and other chronic NCD, or who were taking medication that could affect diet were excluded.

Intervention/Exposure: Studies were included if they used FBDGs to evaluate dietary intake in their own country. Guidelines developed by non-government institutions were excluded. Studies were included if they assess FBDGs through dietary assessment methods, such as food records, 24 h recalls (24hDR) and food frequency questionnaires (FFQ). Studies assessing diet quality and/or adherence to guideline using indexes (e.g., adherence to Dietary Guidelines for Americans using Healthy Eating Index, Alternate Healthy Eating Index, and Dietary Diversity Score) were excluded because they may have assessed additional items outside the FBDGs. Adherence to recommendations in national FBDGS was assessed based on individual level meeting or not meeting the national FBDGs food groups recommendations.

Comparison: Different study designs, i.e., cross-sectional, cohort, and interventions (randomized and non-randomized trials) were included in this review. If intervention design was used, no exclusion criteria were placed on duration, length of follow-up or date.

Outcome: The key outcome of this review was to assess the adherence of participants’ dietary intake to their respective national FBDGs. Studies were excluded if they focused on an outcome other than adherence to a national FBDG, i.e., obesity or other chronic NCD.

A secondary outcome of this review was to assess the difference in adherence to each national FDBG according to their socio-economic classification for each country. Countries were dichotomized into two types of economy: HIC and LMIC; upper-middle, lower-middle, and low-income countries were identified as LMIC [23,24].

### 2.3. Data Extraction

Data were independently extracted from eligible studies by two reviewers (A.L. and S.H.) and cross-checked for accuracy by a third reviewer (J.H.). The extracted data included sample characteristics (age, sex, race/ethnicity, educational level), country, guideline used (name) and adherence to the guideline (reported as mean (± standard deviation/error, SD/SE) or frequency (%)).

### 2.4. Data Synthesis

Due to the heterogeneity of the study population and FBDG features (i.e., focused components, e.g., energy and nutrients vs. other degree of food processing), it was not possible to perform a meta-analysis. A narrative summary of the findings was conducted.

### 2.5. Quality Assessment and Risk of Bias

Study quality was assessed using a designed appraised tool developed by Effective Public Health Practice Project (EPHPP) [25,26] for observational, cross-sectional, before and after studies, and randomized controlled trials. Individual component and overall quality ratings were scores as 1 for strong, 2 for moderate, and 3 for weak.

## 3. Results

### 3.1. Literature Search and Screening

Studies included in this review are summarized in Figure 1. A total of 12,557 eligible papers were identified: 2851 from CIANHL/EBSCO, 411 from Lilacs/SciElo, 1307 from ProQuest, 2413 from PubMed, 4508 from Scopus, and 1052 from Web of Science. After excluding duplicates and reading titles, 2802 studies were assessed for eligibility. Finally, 616 full-text articles met the inclusion criteria and 49 were considered for the qualitative synthesis.

### 3.2. Study Design Characteristics of the High-Income (HIC) and Low- and Middle-Income Countries (LMIC)

From the 49 articles included, only 1 article (2.0%) was case-control [27] while 83.7% of the studies (*n* = 41) were cross-sectional, 4 (8.2%) longitudinal [28,29,30,31], and 3 (6.1%) randomized controlled trial [32,33,34]. From the 41 cross-sectional studies, 21 (42.9%) were performed in representative samples of individuals [35,36,37,38,39,40,41,42,43,44,45,46,47,48,49,50,51]. Table 1 shows details of the studies, which included the FBDG from each country. Thirty-nine studies were conducted in HIC, while the other 10 studies were from LMIC [41,42,51,52,53,54,55,56,57,58].

The average sample size for the HIC studies were 12,355 ranging from 32 [34] to 25,2425 [30], while average sample size for the LMIC were 745,050 ranging from 490 [57] to 32,898 [51]. The studies were conducted in the following countries: United States of America (USA) (*n* = 17) [27,32,34,35,46,48,49,50,59,60,61,62,63,64,65,66], Canada (*n* = 6) [31,36,44,67,68,69], Brazil (*n* = 4) [42,51,53,54], Switzerland (*n* = 4) [29,37,40,43], Australia (*n* = 3) [70,71,72], China (*n* = 2) [55,58], Belgium (*n* = 2) [38,73], Spain (*n* = 2) [30,47], Denmark (*n* = 1) [28], Egypt (*n* = 1) [52], Germany (*n* = 1) [74], Greenland (*n* = 1) [75], Iceland (*n* = 1) [33], Malaysia (*n* = 1) [56], Mexico (*n* = 1) [41], Qatar (*n* = 1) [39], South Korea (*n* = 1) [45], and Sri Lanka (*n* = 1) [57]. Most of the studies (*n* = 32, 65.3%) were conducted in adults ranging from 18 to 65 years old [27,28,29,32,34,35,36,37,39,40,42,43,48,50,51,52,53,54,55,57,59,60,61,64,65,67,70,71], five (10.2%) [30,38,44,62,73] include all age groups, but stratifying them (i.e., children, adolescents and/or adults), six (12.2%) [31,45,58,69,72,75] included only adolescents (10–19 years old), and four (8.2%) [33,49,56,63] included children and/or adolescents (2–19 years).

### 3.3. Adherence to the National Food-Based Dietary Guidelines (FBDGs)

The adherence to the dietary guidelines is reported on Table 2. The majority of the studies reported adherence to the national FBDGs as percentage of meeting the recommendations [28,29,30,35,36,37,38,39,40,41,43,44,46,47,48,51,52,53,56,58,59,61,64,65,68,72,74,75], while only four [35,45,48,70] reported as mean (±SD/SE) servings/day of a certain food group. Almost 40% of the population from both HIC and LMIC do not adhere their national FBDGs [28,29,37,43,44,51,52,56,61,62,63,65,75].

#### Adherence to National FBDGs in HIC and LMIC by Food Groups

The percentages of meeting the guidelines in the HIC ranged from 14.0% [75] in Greenland to 43.0% [65] in the USA, and in the LMIC from 40.0% [56] with Malaysians and 45.0% [52] with Egyptians meeting at least one recommendation from their country-specific FBDG. The food groups that were most frequently reported as having been met for the dietary guidelines were fruits and vegetables (*n* = 19, 38.8%) [27,30,35,37,38,40,41,43,44,48,52,53,56,64,70,72,74,75]. The adherence to fruit group recommendations in HIC varies from 14% in a population of school-age children from Greenland [75] to 67.3% in the overall Spanish population [30], and in LMIC from 13.4% in Malaysian children and adolescents [56] to 49.4% in the overall Brazilian population [53]. The adherence to vegetable group guidelines in HIC varies from 11.4% in the US population [59] to 43.7% in the Spanish population [30], and in LMIC varies from 9.5% in Malaysian adolescents [56] to 74.1% in the Brazilian population [53]. Some HIC and LMIC reported the combined fruit and vegetable guideline adherence: 18% in Swiss population [40], 7–16% in Mexican population [41] and 33.4% in Egyptian adolescents [52]. The adherence reported in HIC was less than 1.5 and 3 servings/day for fruit and vegetables (FV), respectively [48,70]. More specifically, a Canadian study with 33,850 individuals over 2 years reported the adherence to recommendations for dark green vegetables and orange fruits of 12% and 8%, respectively [44].

Dietary intake of discretionary foods, i.e., foods high in fat and oils, sugars and sweets, and sodium, were reported by 11 studies (22.4%) [30,35,36,38,41,51,52,53,70,74,75]. In HIC, studies were mixed in regards to reporting food groups, i.e., one study reported that 36.5% were exceeding the recommendations for sweets [30], 90.1% for oils [74], and 14.7% and 18.6% for cookies and soft-drinks, respectively [75]. The average servings/day for discretionary foods and fats/oils were 1.5 [70]. Studies in LMIC, also varied when reporting the groups of discretionary foods: 99.6% and 58.3% of adults from Brazil exceeded recommendations for fats/oils and sugars [53]; 56.4% and 58.9% of adolescents from Egypt for sweets, fast-food and canned foods [52]; and 84% and 72% of children and adolescents in Mexico for sugar-sweetened beverages and high-saturated fat and added sugars [41]. Louzada et al. [51] showed that only 20.4% of the Brazilian population were eating ultra-processed foods.

Other food groups from national FBDGs reported in the studies included: grains (whole vs. refined), proteins (eggs, meats, and fish), and dairy and alternatives. Reported adherence to guidelines for grains and cereals were 3.8% for women in the USA [64], 79.2% for Greenlandic adolescents consuming potatoes [75] and 50.5% of Germans consuming whole grains [74]. Schwartz and Vernarelli [35] found that individuals that used the MyPlate^®^ guideline to inform eating patterns had higher intakes of whole grains (1.1 to 0.8 servings/day) and refined grains (6.0 to 6.6 servings/day) as compared to those who did not use the guideline. In LMIC, less than 40% population of the met recommendations for adequate grain intake in their national FBDGs, [53,56,58], especially among younger participants. In HIC, adherence to milk and alternatives guidelines ranged from 8.4% in Switzerland [43] to 51.9% in US women [64]; and in LMIC from 5.5% with Malaysian youth [56] to 12.5% in an overall sample of Brazilian adolescents and adults [53]. One Australian [70] and one USA [48] study reported an average of 1.0 serving/day of dairy group. Finally, HIC studies that evaluated the adherence to guidelines for meat and alternatives showed an average intake 30.0% higher than the recommendations [30,37,43,64,74,75], with one study showing 95% of the population meeting the recommendations [74]. Alternatively, the adherence to guidelines for meat and alternatives in LMIC showed mixed results. In Brazil [53] and in China [58] adherence was greater than 65.0%, but in Egypt [52], Malaysia [56], and Mexico [41] it was less than 23.0%. Koo et al. [56] only reported the intake for fish and seafood, and the majority of the studies both from HIC and LMIC reported a high prevalence for (red) meat consumption.

### 3.4. Risk of Bias of the Included Studies

From all the included studies, selection bias (2.2 ± SD 0.5) was the most reported bias, while study design (0.1 ± SD 1.2) the less reported bias. Figure 2 shows the risk of bias of each component rating for the included studies.

## 4. Discussion

This review synthesized the evidence from observational and intervention studies reporting the adherence to national FBDGs in individuals from both HIC and LMIC. The 48 studies included in this review were conducted across 15 HIC and 4 LMIC, thus representing a broad perspective on this study objective. This review found that a large proportion of individuals in both HIC and LMIC are not meeting national dietary guidelines. Meat and alternatives, and discretionary foods were consumed above the recommended amounts, and vegetable intake was below the recommendations. A global review of 90 FBDGs found that most of the dietary guidelines demonstrated that the food groups that were most adhered to were the starchy staples (e.g., rice and potatoes) and the fruit and vegetables, while other groups were less adhered to across the countries [16].

Evidence from this review suggests that the global population may not be meeting the minimum dietary recommendations for FV and whole-grains, with more pronounced deficits in those from HIC. All national FBDGs have common themes that FV and whole grains should be incorporated in a healthy diet for the prevention of obesity, other chronic NCDs, and some nutrient deficiencies [11,12,76]. These food sources usually provide a low amount of fat, and are key sources of vitamins, minerals, and dietary fiber [77]. In some countries, particularly LMIC, where diets of nutritionally vulnerable groups (i.e., children, adolescent girls, women of child-bearing age, and older adults) continue to be inadequate, the co-occurrence of deficiencies from more than one micro-nutrient is common [76,78]. Thus, population-level interventions to improve dietary intake are urgently needed to reverse these global trends.

Findings from this review also demonstrated that discretionary foods and other high sugar and fat sources may exceed the recommendations in national FBDGs. However, it may be noted that most of the included studies assessing fat consumption only reported total fat intake or the combination of fats and oils, with the exception of one study that evaluated food groups and omega-3 fatty acid consumption after a nutrition education program [32]. Thus, most studies did not differentiate between the types of fat consumed in the diet, such as saturated fats, trans fats, and unsaturated fats (e.g., omega 3/omega 6 fatty acids). Given the importance of the associations between different types of fatty acid consumption in different age groups and prevention of several negative health outcomes, such as chronic NCD in adults and older adults [79] and cognitive development in children [80], this information might be valuable. For example, it would be beneficial to distinguish between the negative impacts of saturated/trans fats and positive effects of unsaturated fatty acids in the diet. Furthermore, the finding that the consumption of discretionary foods, such as SSB and sweet snacks, exceed recommendations (i.e., 1 serving = 600kj (143kcal)) [81,82] is crucial for researchers and practitioners given that consumption of these foods may contribute to unhealthy weight gain [83,84]. Unhealthy weight gain is associated with several risk factors for poor health outcomes, public health policies, behavioral-change strategies, and periodical updates on the national FBDGs are needed to tackle this problem.

The studies included in this review suggest that dietary intake patterns differ across age groups. For instance, children and adolescents usually consume fewer FV and other fiber sources [85,86] than adults. Notably, evidence from systematic reviews [87,88] showed that children and adolescents had lower adherence to national FBDGs, and that families play an important role in influencing their eating behaviors [89,90]. Thus, family aspects of eating behavior might be included as key messages on public health initiatives and other resources for the population.

The strengths of this systematic review include the examination of a topic that filled a gap in the existing literature. This systematic review aimed to identify adherence to national FBDGs and verify possible differences between HIC and LMIC. The studies included in this review were conducted in five different continents (North America, South America, Asia, Europe, and Oceania), providing a global perspective on the topic.

This study was not without limitations. Only studies including children older than 2 years, adolescents, and adults were included, as nutrition recommendations for younger children and older adults may differ. Only studies that directly compared intake to their national FBDGs were included; no other measurement tools that were based on the recommendations (e.g., Healthy Eating Indexes) or direct comparisons to international energy and nutrient recommendations (e.g., Institute of Medicine and World Health Organization) were included. This may have caused confirmation bias, when interpreting the studies [91]. Some studies were not performed in representative samples of the correspondent population, compromising their representativeness. Also, the methodology has its own limitations, as, some of the articles included evaluated the dietary intake with one 24hDR with limited items, which is not representative of the habitual diet. Nevertheless, this method is accepted for studying the intake in a large sample of population and estimating the mean nutrients intake [92]. Reported dietary intake may provide biased results (under or over-reporting by participants) due to social desirability. Furthermore, only English publications were included. Additionally, a greater proportion of the studies that evaluate adherence to the FBDGs were from HIC, especially from the USA, limiting the generalizability of the results to other countries. Finally, the selected studies included a variety of population sub-groups (e.g., sex, age, and weight status) that made it difficult to make conclusions across studies.

## 5. Conclusions

National dietary guidelines can be a useful tool to promote a healthy diet for different age groups. A diet based on these guidelines should provide adequate energy and nutrient intake and support a healthy weight status and positive health outcomes. The findings from this review demonstrate that individuals in both HIC and LMIC may be falling short in at least one recommendation from national guidelines. Overall, these results suggest that a substantial proportion of the population are not consuming enough FV and whole grains. Excess intake of discretionary foods was also observed, especially among younger populations from both HIC and LMIC, and the overall population in HIC. Thus, socio-demographic factors (e.g., age, sex, and income) may influence adherence to the guidelines. These findings can help inform the development of future health policies, behavioral-change strategies, and food-based dietary guidelines.

## Figures and Tables

**Figure 1 nutrients-13-01038-f001:**
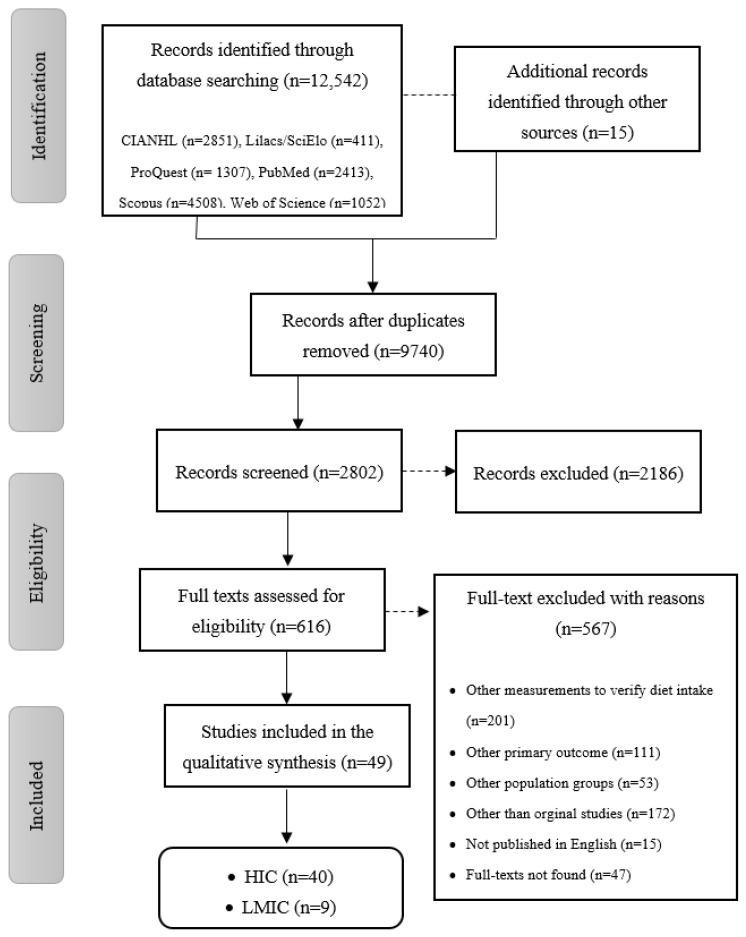
Flowchart of the included studies in the systematic review.

**Figure 2 nutrients-13-01038-f002:**
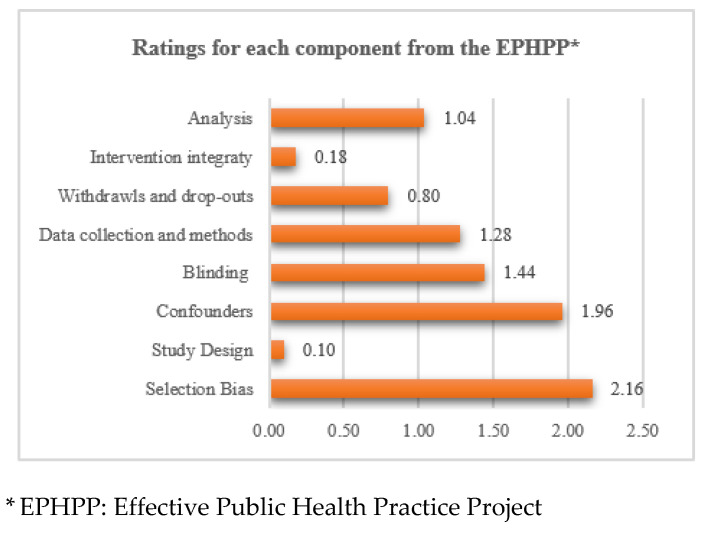
Risk of bias of the included studies based on the Effective Public Health Practice Project: ratings for each component.

**Table 1 nutrients-13-01038-t001:** Study characteristics and adherence to dietary guidelines.

Reference	Study Design	Country	N and Sex (% Female)	Age	Race/Ethnicity	Dietary Measurement	Dietary Guideline
High-Income countries							
Ewers et al., 2020 [28]	Longitudinal (2003–2015)	Denmark	100,191 45.2% female	20–30 y–40–100 y 58.0 ± 13.1	NR	FFQ	Danish Food Based Dietary Guidelines
Schwartz and Vernarelli 2019 [35]	Cross-sectional	USA	3194	≥18 yo	NR	1× 24hR	Dietary Guidelines for Americans/MyPlate or MyPyramid
Schuh et al., 2019 [29]	Longitudinal (1993–2016)	Switzerland	M = 689 50.5% female	35–74: 51.9 yo ± 11 yo	Non-Swiss: 31.4%	FFQ	Swiss Dietary Guidelines
Schroeter et al., 2019 [27]	Case-study	USA	57	19.94 ± 1.20	NR	1× 24hR	Dietary Guidelines for Americans/MyPlate
Irwin et al., 2019 [70]	Cross-sectional	Australia	115 67.8% female	M = 22 ± 5 y	NR	Records	Australian Dietary Guidelines
Harrison et al., 2019 [36]	Cross-sectional	Canada	50.2% female	M = 45.0 ± 0.3 y	NR	1× 24hR	2019 Canada Food Guide
Dıaz-Méndez and Garcıa-Espejo 2019 [30]	Longitudinal (2006–2011–12)	Spain	M = 25,2425	≥16 yo	NR	FFQ	Guide to Healthy Eating/Spanish Society for Community Nutrition
Mestral et al. 2019 [37]	Cross-Sectional	Switzerland	15450 53.0% female	48.8 ± 17.4 y	NR	FFQ	Swiss Dietary Guidelines
Bel et al. 2019 [38]	Cross-Sectional	Belgium	3146	3–64 y: stratified by age	NR	1× 24hR + FFQ	Flemish Active Food Triangle
Schwartz and Vernarelli et al. 2018 [35]	Cross-sectional	USA	3194	18–50+ y	NR	1× 24hR	Dietary Guidelines for Americans/MyPlate or MyPyramid
Jun et al., 2018 [59]	Cross-sectional	USA	3142 100% female	19–70 y	Non-Hispanic, White: 51.8% Non-Hispanic, Black: 19.3% Hispanic: 22.2%	1× 24hR	Dietary Guidelines for Americans/MyPlate
Jones et al. 2018	Cross-sectional	UK	2045 56.5% female	≥18 y	NR	3d food record	UK Government Scientific Advisory Committee on Nutrition
Brassard et al. 2018 [67]	Cross-sectional	Canada	1147 50.2% female	18–65 y	Caucasian: 94.3% African-American: 2.4% Hispanic: 1.7% Other: 1.6%	3× 24hR	2007 Canada Food Guide
Al Thani et al., 2018 [39]	Cross-sectional	Qatar	1109 48.6% female	18–64 y	NR	FFQ	Qatar Dietary Guidelines
Stroebele-Benschop et al. 2018 [74]	Cross-sectional	German	103 75.7% female	18–30: 24.3 ± 3.1 y	NR	FFQ	German Nutrition Society
Chatelan et al., 2017 [40]	Cross-sectional	Switzerland	2086 54.7% female	18–75: 46.8 ± 15.8 y	NR	2× 24hR	Swiss Dietary Guidelines
Mishra et al., 2015 [71]	Cross-sectional	Australia	16227 100% female	18–75: 43.5 ± 1.5 y	NR	FFQ	Australian Guide to Healthy Eating
Schumacher et al., 2014 [72]	Cross-sectional	Australia	332 100% female	13.7: 13.4–13.9 y	Australian: 86% (Aboriginal and Torres Strait Islander: 11%) European: 10% Asian: 1% Other: 3%	FFQ	Australian Guide to Healthy Eating
Yen and Lewis 2013 [32]	Randomized controlled trial	USA	85:41 intervention vs. 44 control 100% female	53.8 ± 6.6 y	Control vs. Intervention White: 95.5 vs. 100% African-American: 2.3% Hispanic: 2.3%	FFQ	Dietary Guidelines for Americans/MyPlate
Abreu et al., 2013 [43]	Cross-sectional	Switzerland	4371 53.8% female	35–75: 57.6 ± 10.5 y	Switzerland: 65.1%	FFQ	Swiss Dietary Guidelines
Black and Billette 2013 [44]	Cross-sectional	Canada	33,850 43.2% female	2–51+ y	French-British Canadian: 57% Other: 43.1%	2× 24hR	2007 Canada Food Guide
Rossiter et al., 2012 [31]	Longitudinal (2002 and 2005)	Canada	247 46.4% female	14–16 y	NR	FFQ	2007 Canada Food Guide
Park et al., 2012 [45]	Cross-sectional	South Korea	394 53.8% female	13.96 ± 0.44 y	NR	1× 24hR	Korean National Dietary Guidelines
McDaniel and Belury, 2012 [60]	Cross-sectional	USA	60 50.0% female	25.5 ± 6.3 y	Non-Hispanic, White: 78.3% Non-Hispanic, Black: 5.0% Asian: 11.7% Indian: 5.0%	FFQ	Dietary Guidelines for Americans, MyPyramid
Kirkpatrick et al., 2012 [46]	Cross-sectional	USA	16,338 adults and children		Non-Hispanic, White: 41.1% Non-Hispanic, Black: 25.8% Mexican-American: 25.3%	1× 24hR	Dietary Guidelines for Americans, MyPyramid
Allen et al., 2011 [68]	Cross-sectional	Canada	291 71.7% female	20–35+ y	NR	Food record	2007 Canada Dietary Guide
Winham and Florian 2010 [61]	Cross-sectional	USA	171 100% female	18–60: 34.4 ± 9.1 y	Hispanic: 76.7% Bicultural or English dominant: 23.4%	FFQ	Dietary Guidelines for Americans, MyPyramid
Niclasen and Schnor 2010 [75]	Cross-sectional	Greenland	2462	11–17 y	NR	FFQ	Greenlandic Board of Nutrition
Kristjansdottir et al., 2010 [33]	Randomized controlled trial	Iceland	106: 58 intervention and 48 control	3–9 y	NR	3d food records	Food-based Dietary Guidelines set for Icelandic population
Kreb-smith et al., 2010 [62]	Cross-sectional	USA	16338	2–71+ y (stratified by age group and sex)	NR	2× 24hR	Dietary Guidelines for Americans, MyPyramid
Vandevijvere et al., 2009 [73]	Cross-sectional	Belgium	3168 43.4% female	≥15 y (stratified by age)	NR	2× 24hR + FFQ	Flemish Active Food Triangle
Kranz et al., 2009 [63]	Cross-sectional	USA	214 45.8% female	2–12 y (stratified age)	Non-Hispanic, White: 47.7% Non-Hispanic, Black: 43.5% Non-Hispanic, Asian: 18.7% Other: 10.3% Hispanic: 14.0%	3× 24hR	Dietary Guidelines for Americans, MyPyramid
John et al., 2008 [69]	Cross-sectional	Canada	1410	Grade 7 and 11 students	NR	1× 24hR + FFQ	2007 Canada Food Guide
Serra-Majem et al., 2007 [47]	Cross-sectional	Spain	2160 53.9% female	NR	NR	2× 24hR + FFQ	Guide to Healthy Eating/Spanish Society for Community Nutrition
Tande et al., 2004 [48]	Cross-sectional	USA	9111 51.4% female	20–59: 37.4 ± 0.2 y	Non-Hispanic, White: 75% Non-Hispanic, Black: 10.4% Mexican-American: 5.9% Others: 8.5%	1× 24hR	Dietary Guidelines for Americans, Food Pyramid
Pullen and Walker, 2002 [64]	Cross-sectional	USA	371 100% female	34–86: 62 y	White: 92.6% Black: 0.5% Hispanic: 2.2% Asian: 3.3% Native American: 0.8% Other: 0.5%	FFQ	Dietary Guidelines for Americans, Food Pyramid
Anding et al., 2001 [65]	Cross-sectional	USA	103 100% female	17–42: 21.6 ± 4.6 y	Black: 23% White: 32% Asian: 20% Hispanic: 23%	3d food records	Dietary Guidelines for Americans, Food Pyramid
Brady et al., 2000 [66]	Cross-sectional	USA	109 56.9% female	7–14:10.2 ± 1.7 y	NR	1× 24hR	Dietary Guidelines for Americans, Food Pyramid
Munõz et al., 1997 [49]	Cross-sectional	USA	3307 50.2% female	2–19: stratified by age	White, Non-Hispanic: 67.9% Black, Non-Hispanic: 16.6% Hispanic: 11.9%	2× 24hR	Dietary Guidelines for Americans, Food Pyramid
Cleveland et al., 1997 [50]	Cross-sectional	USA	8181 58.7% female	20–60+: stratified by age	White, Non-Hispanic:77.1% Black, Non-Hispanic: 11.8% Hispanic: 8.3% Others: 2.8%	1× 24hR + 2 food records	Dietary Guidelines for Americans, Food Pyramid
Gambera et al., 1995 [34]	Randomized- trial	USA	32 37.5% female	33.3 ± 6 y	NR	3d food records	Dietary Guidelines for Americans, Food Pyramid
Low- and middle-income countries							
Steele et al., 2020 [54]	Cross-sectional	Brazil	10,116 78.0% female	18–60+ y: stratified by age	NR	FFQ	Dietary Guideline for the Brazilian Population 2014
Ansari and Samara 2018 [52]	Cross-sectional	Egypt	2422 53.8% female	18.9 ± 1.4 y	NR	FFQ	WHO dietary guidelines for Eastern Mediterranean region
Sousa and Costa 2018 [53]	Cross-sectional	Brazil	506 57.0% female	20–50+ y: stratified by age	NR	2× 24hR	Brazilian Dietary Guideline 2006/Food Pyramid
Louzada et al., 2018 [51]	Cross-sectional	Brazil	32,898	≥19 y	NR	2× 24hR	Dietary Guideline for the Brazilian Population 2014
Tian et al., 2017 [55]	Cross-sectional	China	14,452 51.9% female	20–59: 42.8 ± 10.3 y	NR	3× 24hR	Chinese Food Pagoda
Batis et al.., 2016 [41]	Cross-sectional	Mexico	7983 50.6% female	5–20+ y: stratified by age	NR	1× 24hR	Mexican Dietary Guidelines
Chin Koo et al., 2016 [56]	Cross-sectional	Malaysia	1773 48.6% female	7–12: stratified by age	Malay: 59.2% Chinese: 19.5% Indian: 6.7% Others: 14.7%	FFQ	Malaysian Dietary Guidelines
Verly-Jr et al. 2013 [42]	Cross-sectional	Brazil	1661 56.5% female	37.7 ± 29.9 y	NR	2× 24hR	Dietary Guidelines for Brazilian population/Pyramid
Jayawardena et al., 2013 [57]	Cross-sectional	Sri Lanka	490 65.5% female	48.3 ± 15.6 y	Sinhalese: 75.6% Muslim: 5.9% Sri Lankan Tamil: 9.5% Indian Tamil: 9%	2× 24hR	Food-based Dietary Guidelines for Sri Lanka and Dietary Guidelines for Brazilian population/Pyramid
Zhang et al., 2012 [58]	Cross-sectional	China	2204 53.8% female	12–17: 15.1 ± 1.9 y	NR	FFQ	Chinese Food Pagoda

24hR: 24 h recall; M = mean; FFQ: Food Frequency Questionnaire, NR: Not Reported; Y = years. Note: Countries were identified as high and low- to middle-income countries based on the World Bank criteria.

**Table 2 nutrients-13-01038-t002:** Population adherence to the dietary guidelines across different high- and low- to middle-income countries.

Reference	Dietary Guidelines	Pictorial Image	Food Groups	Include Physical Activity Messages	Summary Results
High-income countries					
Ewers et al., 2020 [28]	Danish Food Based Dietary Guidelines	No image Key messages	Fruits and vegetables Fish Whole Grains Lean meats and lean cold meats Low-fat dairy products Saturated fat sources Sodium sources Sugar Water	Yes	Only 10% of the participants were identified as very high adherence; 17.3% high adherence; 54.4% intermediate adherence; 8.3% low adherence; and 9.9% very low-adherence. Based on table categorization of meeting the guideline proposed by the authors.
Schwartz and Vernarelli 2019 [35]	Dietary Guidelines for Americans	Plate and Pyramid	Fruits Vegetables Grains Dairy Protein Foods	No	Following MyPlate and MyPyramid showed better adherence to the recommendations than those who did not follow. More whole grains (1.1 vs. 0.8 servings), and vegetables (1.5 vs. 1.4 servings) Less refined grains (6 vs. 6.6 servings) and added sugar (18.6 vs. 20.5 tbs) sources
Schroeter et al., 2019 [27]	Dietary Guidelines for Americans	Plate and Pyramid	Fruits Vegetables Grains Dairy Protein Foods	No	Increased consumption of food groups after participating in education groups Fruit and vegetables and whole grains
Schuh et al., 2019 [29]	Swiss Dietary Guidelines	Pyramid	Beverages Vegetables and Fruits Grains, potatoes and pulses Dairy products, meat, eggs, fish and tofu Oils, fats, and nuts Sweets, salty snacks, alcohol	Yes	Participants are not meeting the guidelines five year after issuing the guideline, regardless of socio-demographic characteristics. Meeting at least three recommendations 1993: 26.1% 2006: 24.9%
Irwin et al., 2019 [70]	Australian Dietary Guidelines	Plate	Grains (mostly whole grains) Vegetables and legumes/beans Lean meats, and poultry, fish, eggs, tofu, nuts and seeds, and legumes/beans Milk, yogurt, cheese and/or alternatives mostly reduced fat Fruit Use small amounts of oils and fats Only sometimes and small amounts: sugar sweetened beverages, salty snacks and sweets.	No	Participants are not meeting the guidelines for the majority of the food groups Lean meats, and alternatives were the only group that students were meeting the recommendations Meeting the recommendations (female vs. male): Fruit: 0.7 vs. 1.3 servings/day Vegetables: 2.7 vs. 3.2 Meat and Alternatives: 3.0 vs. 2.2 Dairy and Alternatives: 1.3 Bread, cereals, grains: 4.3 vs. 3.3 Discretionary: 1.3 vs. 2.0 Fats and oils: 1.7 vs. 1.0
Harrison et al., 2019 [36]	2019 Canada Food Guide	Plate	Vegetables and Fruits Protein foods Whole grains Water	No	Greater consumption of saturated fats from all the food groups of the Canada Food Guide Protein Foods (milk and alternatives; and meats and alternatives) contributed 47.8% in total for saturated fats. “All other foods” were main contributors: fruit juices, refined grains, and salty snacks.
Diaz-Mendez and García-Espejo 2019 [30]	Guide to Healthy Eating–Spanish Society for Community Nutrition	Pyramid	Whole grains Fruits Vegetables and legumes Oils (especially olive oils) Lean meats, poultry, fish, eggs Beans and nuts Milk and dairy Water	Yes	Percent of participants meeting the guidelines: Fruits: 67.3% Vegetables: 43.7% Meat: 52.3% Breads: 86.6% Percent of participants not meeting the guidelines: Eggs: 59.3% Fish: 46.4% Pasta-rice-potatoes: 47.3% Sweets: 36.5%
Mestral et al., 2019 [37]	Swiss Dietary Guidelines	Pyramid	Beverages Vegetables and Fruits Grains, potatoes and pulses Dairy products, meat, eggs, fish and tofu Oils, fats, and nuts Sweets, salty snacks, alcohol	Yes	Less than 40% of the participants adhere to all of the guidelines Percent of participants meeting the guidelines: Fruits: 38.8% Vegetables: 20.5% Dairy: 19.4% Fish: 22.5% Meat: 9.1% Liquids (beverages): 39.4%
Bel et al., 2019 [38]	Flemish Active Food Triangle	Triangle	Cereals and Potatoes Vegetables Fruits Meat, fish, eggs, and meat alternatives Dairy and calcium-enriched products Oils and fatty products Sugary products Unsweetened beverages (water and tea)	Yes	Between years the participants adherence to guidelines deteriorated over the time for most groups Change in percent of participants meeting the guidelines: Water and sugar-free drinks: △+7% Bread and cereals: △−15% Potato, rice, and pasta: △+2% Vegetable: △−1% Fruit (including juices): △−3% Fruit (excluding juices): △−1% Dairy products and calcium-enriched products: △−1% Cheese: △+4% Meat, eggs, fish, and substitutes: △−2% Spreadable and cooking fat: △−4%
Schwartz and Vernarelli, 2018 [35]	Dietary Guidelines for Americans	MyPlate	Fruits Vegetables Grains Dairy Protein Foods	No	Participants who follow a MyPlate plan were able to meet the food groups requirements for the following groups: Dark green and orange vegetables Refined grains Whole grains Total meat Milk and dairy Sodium sources
Jun et al., 2018 [59]	Dietary Guidelines for Americans	MyPlate	Fruits Vegetables Grains Dairy Protein Foods	No	A small percentage of participants met the dietary guidelines: Fruits: 21.3% Vegetables: 11.4% Whole grains: 4.3%
Brassard et al., 2018 [67]	2007 Canada’s Food Guide	Rainbow	Vegetables and fruits Grain products Milk and alternatives Meat and alternatives	No	Participants were consuming less than the recommendations for fruits and vegetables, and grain-products. Milk and alternatives; and Meat and alternatives are in line with the recommendations.
Stroebele-Benschop et al., 2018 [74]	German Dietary Guidelines	Circle	Cereals and potatoes Vegetables Fruits Milk and dairy products Meat, sausages, fish, and eggs Fats and oils	No	Participants were not meeting the recommendations for most of the food groups. Percentage of participants that met the recommendations: Vegetables: 12.9% Fruit: 37.6% Whole grain: 50.5% Milk and milk products: 45.5% Meat and meat products: 95.0% Fish: 15.8% Eggs: 80.2% Oil: 90.1% Fat: 89.1% Water and unsweetened beverages: 76.2%
Chatelan et al., 2017 [40]	Swiss Dietary Guidelines	Pyramid	Beverages Vegetables and Fruits Grains, potatoes and pulses Dairy products, meat, eggs, fish and tofu Oils, fats, and nuts Sweets, salty snacks, alcohol	Yes	Less than 1% follow all the food groups. Percentage of participants that met the recommendations: Fruit and vegetables: 18.0% Non-caloric beverages: 75.0%
Mishra et al., 2015 [71]	Australian Dietary Guidelines	Plate	Grains (mostly whole grains) Vegetables and legumes/beans Lean meats, and poultry, fish, eggs, tofu, nuts and seeds, and legumes/beans Milk, yogurt, cheese and/or alternatives mostly reduced fat Fruit Use small amounts of oils and fats Only sometimes and small amounts: sugar sweetened beverages, salty snacks and sweets.	No	Younger age participants were not meeting the recommendations for all the food groups. Middle-age participants were not meeting the recommendations for cereals, vegetables, and meat and alternatives.
Schumacher et al., 2014 [72]	Australian Dietary Guidelines	Plate	Grains (mostly whole grains) Vegetables and legumes/beans Lean meats, and poultry, fish, eggs, tofu, nuts and seeds, and legumes/beans Milk, yogurt, cheese and/or alternatives mostly reduced fat Fruit Use small amounts of oils and fats Only sometimes and small amounts: sugar sweetened beverages, salty snacks and sweets.	No	Participants were meeting the recommendations for the guideline for the majority of the groups. Fruit: 23.8% Vegetables: 28.6% Dairy: 15.7% Breads and Cereals: 5.7% The only exception was for the meat and substitutes group, where 69.3% were meeting them.
Yen and Lewis 2013 [32]	Dietary Guidelines for Americans	MyPlate	Fruits Vegetables Grains Dairy Protein Foods	No	After participating in an educational program, participants did not improve their intake for the groups: grains, vegetables, and meat and alternatives. They improved their intake for fruit, dairy, and oil.
De Abreu et al. 2013 [43]	Swiss Dietary Guidelines	Pyramid	Beverages Vegetables and Fruits Grains, potatoes and pulses Dairy products, meat, eggs, fish and tofu Oils, fats, and nuts Sweets, salty snacks, alcohol	Yes	Only 23% of the sample were meeting at least 3 recommendations. Fruits: 39.4% Vegetables: 7.1% Dairy: 8.4% Majority were meeting the recommendations for meat and fish. Meat: 61.3% Fish: 66.4%
Black and Billette, 2013 [44]	2007 Canada’s Food Guide	Rainbow	Vegetables and fruits Grain products Milk and alternatives Meat and alternatives	No	Only 26.3% of participants met all of the recommendations. Dark green: 12% Orange fruit: 8% Potatoes: 10% Other: 43%
Rossiter et al., 2012 [31]	2007 Canada’s Food Guide	Rainbow	Vegetables and fruits Grain products Milk and alternatives Meat and alternatives	No	Participants were consuming below the recommendations for fruits and vegetables, and grains. Milk and dairy were consumed in the recommendation range. Meat and alternatives were consumed above the recommendations.
Park et al., 2012 [45]	Korean National Dietary Guidelines	Wheels	Fruits Vegetables Meat, fish, eggs, and beans Milk Grains	Yes	Overall adherence was 3.23 (1–5 Likert scale) for meeting the recommendations.
Kirpatrick et al., 2012 [46]	Dietary Guidelines for Americans	MyPlate	Fruits Vegetables Grains Dairy Protein Foods	No	Over 50% of adults met the recommendations for total grains, meats, and beans; less then 20% of adults met the recommendations for other groups
Mc Daniel and Belury 2012 [60]	Dietary Guidelines for Americans	MyPlate	Fruits Vegetables Grains Dairy Protein Foods	No	Participants’ intakes of fruit and vegetables were below the recommendations. Meat and beans; and Milk and dairy were within the recommendations. Grains were above the recommendations Oils were below the recommendations.
Allen et al., 2011 [68]	2007 Canada’s Food Guide	Rainbow	Vegetables and fruits Grain products Milk and alternatives Meat and alternatives	No	Less than 48% of the sample were meeting the recommendations for all groups (with females having a better adherence then males) Male vs. Female Vegetables and fruits: 9.5% vs. 17.9% Milk and alternatives: 16.2% vs. 22.2% Grain products: 16.2% vs. 38.2%
Winham and Florian, 2011 [61]	Dietary Guidelines for Americans	Pyramid	Fruits Vegetables Grains Milk Meat	Yes	Less than 30% of Hispanics adhere to the guidelines 7% of bi-racial group adhere to the guidelines.
Niclasen and Schnor, 2011 [75]	Greenlandic (similar to the Danish guidelines)	No image	Fruits Vegetables Traditional foods Whole grains Fat Sugar Water	Yes	Students meeting the guidelines varied from 14% to 87% depending on the groups. Diet variety: 87.0% Marine animals: 31.6% Local terrestrial animals and birds: 37.1% Fish: 31.8% Fruit: 14.8% Vegetables: 38.9% Potatoes: 79.7% Candies: 14.7% Soft-drinks: 18.6% Pre-cooked dinner: 83.4%
Kristjansdottir et al., 2010 [33]	Food-based dietary Guidelines set for Icelandic Population	Circle/Plate	Fruits and vegetables Cereals and cereal products Dairy products Animal source foods and nuts Oils and visible fats Water	Yes	After the intervention there was an improvement on the food groups intake, however, the participants from both intervention and control groups were still not meeting the recommendations.
Krebs-Smith et al., 2010 [62]	Dietary Guidelines for Americans	Pyramid	Fruits Vegetables Grains Milk Meat	Yes	Majority of the population did not meet the recommendations for all the food groups, with exception for total grains, and meat and beans.
Vandevijvere et al., 2009 [73]	Flemish Active Food Triangle	Triangle	Cereals and Potatoes Vegetables Fruits Meat, fish, eggs, and meat alternatives Dairy and calcium-enriched products Oils and fatty products Sugary products Unsweetened beverages (water and tea)	Yes	Population was consuming below the recommendations for liquids, grains, vegetables, fruits, and milk and soya products. Population was consuming above the recommendations for meats/fish/eggs/legumes/nuts/substitutes
Kranz et al., 2009 [63]	Dietary Guidelines for Americans	Pyramid	Fruits Vegetables Grains Milk Meat	Yes	Younger children presented a higher adherence to the guidelines than older children. Older children have less than 40% adherence for fruit and vegetables.
St. John et al., 2009 [69]	2007 Canada’s Food Guide	Rainbow	Vegetables and fruits Grain products Milk and alternatives Meat and alternatives	No	Fruit and vegetables are the groups for which the children have the lowest adherence. Milk and dairy; and meat and alternatives have been less adherent to some specific subgroups: normal weight and overweight.
Serra-Majem et al., 2008 [47]	Guide to Healthy Eating–Spanish Society for Community Nutrition	Pyramid	Whole grains Fruits Vegetables and legumes Oils (especially olive oils) Lean meats, poultry, fish, eggs Beans and nuts Milk and dairy Water	Yes	Majority of the population were not meeting the recommendations for fruits (72.7%), vegetables (57.6%), and beans (58.1%). Majority of the population exceeded recommendation for intake of fatty meats and sausages (56.1%). Intake for bake goods was 20.2%, soft-drinks 21.8%, and fats 23.6%, and sugars 33.5%.
Tande et al., 2004 [48]	1992 Dietary Guidelines for Americans	Pyramid	Bread, Cereal, Rice, Pasta Fruit Vegetable Meat, Poultry, Fish, Dry Beans, Eggs, and Nuts Milk, Yogurt, Cheese Fats and oils Sweets	No	Participants were not meeting the recommendations for all the groups, consuming fewer than the recommended. Serving/1000 kcal Dairy: 0.95 Fruit: 0.70 Vegetables: 1.55 Grain: 3.19 Meat: 1.06
Pullen and Walker, 2002 [64]	1992 Dietary Guidelines for Americans	Pyramid	Bread, Cereal, Rice, Pasta Fruit Vegetable Meat, Poultry, Fish, Dry Beans, Eggs, and Nuts Milk, Yogurt, Cheese Fats and oils Sweets	No	Women are not adhering the recommendations for the pyramid groups, with exception for fruits that have adherence of 65.4%. Meat: 38.6% Dairy: 48.1% Vegetables: 22.3% Grain products: 3.8%
Anding et al., 2001 [65]	1992 Dietary Guidelines for Americans	Pyramid	Bread, Cereal, Rice, Pasta Fruit Vegetable Meat, Poultry, Fish, Dry Beans, Eggs, and Nuts Milk, Yogurt, Cheese Fats and oils Sweets	No	Less than 43% of the participants followed at least one recommendation from the guideline. Fruits, vegetables, and milk were consumed less than the recommendations. More than 60% of the participants exceeded the recommendations for fats, sugar and sodium.
Brady et al., 2000 [66]	1992 Dietary Guidelines for Americans	Pyramid	Bread, Cereal, Rice, Pasta Fruit Vegetable Meat, Poultry, Fish, Dry Beans, Eggs, and Nuts Milk, Yogurt, Cheese Fats and oils Sweets	No	Participants from both sexes and ethnicities were consuming less than the recommendations for fruits and dairy. Participants also exceeded the recommendations for sugar and discretionary foods.
Muñoz et al., 1997 [49]	1992 Dietary Guidelines for Americans	Pyramid	Bread, Cereal, Rice, Pasta Fruit Vegetable Meat, Poultry, Fish, Dry Beans, Eggs, and Nuts Milk, Yogurt, Cheese Fats and oils Sweets	No	Participants were not meeting (consuming less) the recommendations for fruit, vegetables, and grains. Dairy and meat intake met the recommendations.
Cleveland et al., 1997 [50]	1992 Dietary Guidelines for Americans	Pyramid	Bread, Cereal, Rice, Pasta Fruit Vegetable Meat, Poultry, Fish, Dry Beans, Eggs, and Nuts Milk, Yogurt, Cheese Fats and oils Sweets	No	Participants were not meeting the recommendations (consuming less) for grains, dairy and fruits. Vegetables and meats were within the recommendations.
Gambera et al., 1995 [34]	1992 Dietary Guidelines for Americans	Pyramid	Bread, Cereal, Rice, Pasta Fruit Vegetable Meat, Poultry, Fish, Dry Beans, Eggs, and Nuts Milk, Yogurt, Cheese Fats and oils Sweets	No	After intervention participants increased their intake for milk, vegetables, fruits, and grains, and decrease intake of meats.
Low- and middle-income countries					
Steele et al., 2020 [54]	Dietary Guidelines for the Brazilian population 2014	No	Unprocessed or minimally processed foods Processed culinary ingredients Processed foods Ultra-processed foods	No	Slight increase in consumption of unprocessed/minimally processed foods after coronavirus disease 2019 (COVID-19) pandemic. Consumption of ultra-processed foods remains the same after COVID-19.
Sousa and Costa, 2018 [53]	Dietary Guidelines for the Brazilian population 2006	Pyramid	Rice, bread, pasta, potato, cassava Fruits Vegetables Beans and nuts Milk, cheese, and yogurt Sugar and sweets Oils and fats	No	Participants were not meeting the guidelines for most of the food groups: Grains: 95.8% Vegetables: 74.1% Fruits: 49.4% Meat: 97.2% Beans: 93.5% Dairy products: 87.5% Fats and oils: 99.6% Sugars: 58.3%
Ansari and Samara 2018 [52]	WHO Dietary Guidelines for the Eastern Mediterranean region	Plate	Bread, cereals, potatoes, and rice Milk and dairy products Foods containing fat Foods/drinks containing sugar Meat, poultry, fish, dried beans, and eggs Fruit and vegetables	Yes	Participants had an adherence for most of the food groups below 45%, exception for cereal/cereal products that had an adherence of 71.8%. Sweets: 43.5% Cakes/Cookies: 44.2% Snacks: 33.0% Fast food/canned foods: 41.1% Lemonade/soft-drinks: 43.7% Fruits and vegetables: 33.4% Dairy products: 19.1% Meat/sausage products: 16.5% Fish/seafood: 32.1%
Louzada et al., 2018 [51]	Dietary Guidelines for the Brazilian population 2014	No	Unprocessed or minimally processed foods Processed culinary ingredients Processed foods Ultra-processed foods	No	Consumption of unprocessed/minimally processed foods was higher than the other groups: Prevalence of intake: Minimally or unprocessed foods: 58.1% Processed culinary ingredients: 10.9% Processed foods: 10.6% Ultra-processed foods: 20.4%
Tian et al., 2017 [55]	Chinese Food Pagoda	Pagoda	Water Cereals and tubers Vegetables and fruits Meat and Poultry; Aquatic products; Eggs Milk and dairy products; soybeans and nuts; Salt and Oils	Yes	Participants were consuming oils and fats above the recommendations, while fruits, eggs, aquatic products, and milk were below recommendations. Vegetables were the only group that meet the recommendations.
Batis et al., 2016 [41]	Mexican Dietary Guidelines	Plate	Fruit and vegetables Cereals Beans and animal protein sources	No	Low proportion adheres to the recommendations. Legumes: 1–4% Seafood: 4–8% Fruit and vegetables: 7–16% Dairy: 9–23% Sugar-sweetened beverages: 10–22% High saturated fat and added sugar products: 14–42% Processed meat: 7–23%
Chin Koo et al., 2016 [56]	Malaysian Dietary Guidelines	Pyramid	Rice, cereals, noodles, and tubers Vegetables and fruits Animal source of foods and legumes Fats, sugar and salt	No	Average consumption of the guidelines was below 40% for most of the food groups, with exception for “meat/poultry” with 84.8% Cereals/grains: 40.1% Fruits: 13.4% Vegetables: 9.5% Fish: 24.7% Legumes: 8.9% Milk/Dairy products: 5.5%
Verly Jr. et al., 2013 [42]	Dietary Guidelines for the Brazilian population 2006	Pyramid	Rice, bread, pasta, potato, cassava Fruits Vegetables Beans and nuts Milk, cheese, and yogurt Sugar and sweets Oils and fats	No	Participants were not meeting the recommendations for grains, fruits, vegetables, meat, oils, and sugars food groups. Participants were meeting the recommendations for milk and dairy, and beans/nuts groups.
Jayawardena et al., 2013 [57]	Food-Based Dietary Guidelines for Sri Lanka	Pyramid	Rice, bread, other cereals, and yams Fruits Vegetables Milk and/or Milk products Fish, pulses, meat, and eggs Nuts, oils, and seeds	No	Participants were exceeding the recommendations for grains, meats and pulses, and added sugars sources. Below recommendations for fruits, vegetables, and dairy.
Zang et al., 2012 [58]	Chinese Food Pagoda	Pagoda	Water Cereals and tubers Vegetables and fruits Meat and Poultry; Aquatic products; Eggs Milk and dairy products; soybeans and nuts; Salt and Oils	Yes	Majority of the participants were not meeting the recommendations for the following groups: Only 6.1%, 1.6% and 3.6% consumed the minimum recommendations for cereals, fruits, and vegetables. Participants consuming more than maximum recommendations for meats (65.4%). Dairy (67.4%), eggs (63.9%), and fish and shrimps (81.8%) were consumed less than the minimum recommendations.

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
