# Peer review of "Adherence to Food-Based Dietary Guidelines: A Systemic Review of High-Income and Low- and Middle-Income Countries"

_nutrients, 2021, doi:10.3390/nu13031038_

Round 1
Reviewer 1 Report
Dear Authors,
The manuscript presents valuable review on the impact of FBDGs on dietary patterns and intake so far on the global level. Considering differences in studies' design, it manages to draw solid but limited conclusions on where to go further with development of health and nutrition public strategies, interventions and FBDGs whatsoever.
Here are some comments on the manuscript:
Line 12: I think that saying different socio-economic countries is not correct expression. Try with "across countries with different socio-economic status".
Line 16: used -> using /applying
Line 17: own country -> national FBDGs
Line 18: What do You mean by the most reported? Reported where? Refine the meaning of the sentence.
Line 22: READ meat -> red meat
Abstract has 222 words. The abstract should be a total of about 200 words maximum.
Line 31: ...million adults are overweight, and 671 million are obese
Line 32-33: number of cases in LMIC are rapidly reaching those observed in HIC
Please, refine spacing between the words in a sentence and reference number in whole manuscript and in the table 2.
Line 42-44: This sentence should be better connected with the following one. " For instance, results from multi center cross-sectional study, assessing diet quality of individuals...for eight Latin American countries (the Latin American Health Study of Nutrition and Health-ELANS) showed that better scores for healthy eating were found in higher socio-economic populations, while scores for unhealthy diet were observed in lower socio-economic populations."
Line 52: In line 108 you introduce the abbreviation again. Please correct there.
Line 68: Again used term "different socio-economic countries"...Please correct.
Line 213-225: Please refine spacing between numbers in brackets and references.
Line 259: Consider introducing 24HDR as an abbreviation for 24 dietary recall
Comment of chapter 3.3: It is not clear what is the point in this chapter. Does it reviews the content of FBDGs and based on what are they created or it reviews dietary methods that were used to assess adherence to the FBDGs? Please, reconsider the entire chapter.
Also, if You reviewed the content of all of the available FBDGs, You should have define it in the objectives of the work.
Line 382: Use abbreviation for high income countries
Line 383: Use only abbreviation for low and middle income countries; use digits for forty-eight
Line 384: what issues? it is not clear on what you refer.
Line 385: their own country's -> national
Line 386: discretionary sources of food-> discretionary foods
Line 392: FV abbreviation could be introduced earlier in the text.
Line 398: child-bearing age
Lien 401: discretionary foods and other high sugar and fat sources
Line 407: omega-6 f.a.???
Line 421-422: I think it is not necessary to bold part of the text.
Line 428: start new paragraph with limitations of the study
Line 439-440: FFQ is an adequate tool for assessment of dietary patterns.
Line 452: their country's guidelines ->national FBDGs
Author Response
Reviewer 1:
Line 12: I think that saying different socio-economic countries is not correct expression. Try with "across countries with different socio-economic status".
Response: Thank you for your suggestion. We made the changes.
Line 16: used -> using /applying
Response: We agree and made the change.
Line 17: own country -> national FBDGs
Response: Thank you and change made.
Line 18: What do You mean by the most reported? Reported where? Refine the meaning of the sentence.
Response: We agree with you. I meant that FV were the food groups with higher adherence shown in the study. I made the changes in the abstract.
Line 22: READ meat -> red meat
Response: We are sorry for that. We correct our typo error.
Abstract has 222 words. The abstract should be a total of about 200 words maximum.
Response: Thank you for your concern in this matter. We reduced the abstract and we have now 193 words.
Line 31: ...million adults are overweight, and 671 million are obese
Response: Thank you we provided the changes.
Line 32-33: number of cases in LMIC are rapidly reaching those observed in HIC
Response: Thank you. We made the change.
Please, refine spacing between the words in a sentence and reference number in whole manuscript and in the table 2.
Response: We apologize for this issue. We provide the changes.
Line 42-44: This sentence should be better connected with the following one. " For instance, results from multi center cross-sectional study, assessing diet quality of individuals...for eight Latin American countries (the Latin American Health Study of Nutrition and Health-ELANS) showed that better scores for healthy eating were found in higher socio-economic populations, while scores for unhealthy diet were observed in lower socio-economic populations."
Response: We agree and added your suggestion for improvement.
Line 52: In line 108 you introduce the abbreviation again. Please correct there.
Response: Thank you. We made the correction in regards to abbreviation.
Line 68: Again used term "different socio-economic countries"...Please correct.
Response: Thank you. We are now using the term suggested.
Line 213-225: Please refine spacing between numbers in brackets and references.
Response: We apologize for this. We correct spacing between numbers in brackets and references in the whole manuscript.
Line 259: Consider introducing 24HDR as an abbreviation for 24 dietary recall
Response: Thank you for your suggestion. We introduced 24hDR as an abbreviation for 24 dietary recall.
Comment of chapter 3.3: It is not clear what is the point in this chapter. Does it reviews the content of FBDGs and based on what are they created or it reviews dietary methods that were used to assess adherence to the FBDGs? Please, reconsider the entire chapter.
Response: Thank you and we deleted the chapter 3.3.
Also, if You reviewed the content of all of the available FBDGs, You should have define it in the objectives of the work.
Response: We agree with you. We added on the objective of the study that we review all the available FBDGs.
Line 382: Use abbreviation for high income countries
Response: Abbreviation for high-income countries were added; i.e., HIC.
Line 383: Use only abbreviation for low and middle income countries; use digits for forty-eight
Response: Thank you we added only abbreviation for low-middle-income countries (LMIC). We use digits for forty-eight – 48.
Line 384: what issues? it is not clear on what you refer.
Response: We apologize for the misunderstanding. We change the word “issues” for “this study objective”.
Line 385: their own country's -> national
Response: Thank you. We changed for national FBDGs …
Line 386: discretionary sources of food-> discretionary foods
Response: Our mistake. We changed for discretionary foods.
Line 392: FV abbreviation could be introduced earlier in the text.
Response: Thank you for your suggestion. We introduced earlier in the text.
Line 398: child-bearing age
Response: Thank you. We added age.
Line 401: discretionary foods and other high sugar and fat sources
Response: Thank you. We made the changes.
Line 407: omega-6 f.a.???
Response: Sorry for that, but we did not understand what was meant by your question on this topic. Actually, it was study that evaluate the adherence to US Dietary Guidelines/MyPlate and sources of omega 3 in middle-age women. Therefore, we thought it would be useful to explain the differences in regard to different food sources of fats (i.e., saturated fats and unsaturated fats)
Yen W-JJ, Lewis NM (2013) MyPyramid-omega-3 fatty acid nutrition education intervention may improve food groups and omega-3 fatty acid consumption in university middle-aged women. Nutrition research 33 (2):103-108. doi: http://dx.doi.org/10.1016/j.nutres.2012.11.015
Line 421-422: I think it is not necessary to bold part of the text.
Response: Our mistake. We removed the bold.
Line 428: start new paragraph with limitations of the study
Response: Thank you for your suggestion. We started a new paragraph.
Line 439-440: FFQ is an adequate tool for assessment of dietary patterns.
Response: We appreciate your comment. We provided the changes.
Line 452: their country's guidelines ->national FBDGs
Response: Thank you. We made changes.
Reviewer 2 Report
Thank you for conducting this work, please find my comments attached.

Author Response
Abstract
Line 11: capitalize Research.
Response: Thank you. We capitalize “Research”.
Lines 14: add ‘A’ – “A systematic review…”
Response: Thank you. We added “A”.
Line 17: consider changing to “Findings indicate that almost 40% of populations…”
Response: We appreciate your comment. We provided your consideration.
Line 18: What do you mean that FV were the most reported? The most reported guideline?
Response: We were sorry for the misunderstanding. We meant that FV was the most reported on the guideline for adherence. Hence, we provided changes.
Line 19: Again, what do you mean by reported for Discretionary? The aim of the analysis is adherence to guidelines, I am not sure the connection being made here to reported.
Response: Thank you for your comment. We made the necessary changes.
Line 21-22: can you quantify animal protein exceeding recommendations here?
Response: We are thankful for the commentary. We provided the approximate quantity in (…).
Introduction
Line 38: I think a word is missing… what is development of flow goods?
Response: We agree. We provided changes to the sentence as you suggested in the next commentary.
Lines 37-40: re-order to begin with the important piece: that the sharing of cultural ideas is increasing. “The sharing of cultural ideas is enhanced with the rapid development…”
Response: Thank you. We re-order this sentence.
Line 40: make this a new paragraph focused on diet quality.
Response: We made a new paragraph.
Lines 42-45: Again, lead with the important piece you want the reader to take away. “Higher diet quality scores were found in higher socio-economic populations while lower diet quality scores were found in lower socio-economic populations (? Is this right?) in the Latin American Health Study of Nutrition and Health, a multi-center cross-sectional study, aimed to assess diet quality of 43 individuals (n=9218) from eight Latin American countries.”
Response: Thank you and this was also commentary from reviewer 1. Thus, we made the changes.
Do you need to abbreviate ELANS? You don’t use the abbreviation in the remainder of the manuscript. Please check your manuscript and make sure you are stating abbreviations when appropriate (ex. Need to state what the US abbreviation is) and not including when the abbreviation is only mentioned once.
Response: We agree with you. We delete the abbreviation and the name of the study.
Line 48: you state diverse, but you haven’t mentioned diet diversity yet.
Response: Thank you. We remove the word diverse.
Line 49: connect these components with diet quality. Instead of optimal, consider “A high quality diet consists of…”.
Response: Thank you. We provided changes.
Line 53-55: Can be more concise in your writing (here and in general). Consider: “FBDGs provide individuals dietary advice to promote health, prevent diet-related diseases.”
Response: We agree with you. We were more concise on this (and tried our best to be concise in other parts of the manuscript).
Lines 59-62: Basically the gap in the literature is that current reviews do not measure adherence to FBDGs, correct? If so, please make this more concise.
Response: Yes, you are correct. We made the changes
Lines 70-72: combine with previous paragraph.
Response: We combine the previous paragraph.
Materials and Methods
Need more information on how adherence was determined!
Response: We apologize for that. We added an explanation on the methods explaining how the adherence was determined.
Line 80: Not sure if ‘the’ is needed?
Response: We removed the wording “the”.
Line 100: Thank you for stating why you choose the older adult ages. Why was the age of 7 years chosen?
Response: We sorry for this and this was our mistake. Our inclusionary criterion was children, adolescents and adults. Only children older than 7y appeared on our systematic review searchers. We corrected this on the methods section. Also, on the limitations of the manuscript, we mentioned that we were limited on the other age groups (due to gap on studies evaluating adherence to FBDGs).
Line 113: I don’t understand why indices would be excluded. The HEI is a standard method used to assess adherence to the Dietary Guidelines for Americans. Is it that adherence isn’t directly assessed with the composite index score?
Response: As we stated on the methods section, we opt not use indexes because although they are based on national FBDGs, they target other additional components not used in the FBDGs.
Line 84 & Line 129 and 130: please be consistent as to whether you put ‘.’ in initials.
Response: We apologize for inconsistencies and opted to remove the ‘’.
Results
Can you more concisely communicate the results (maybe add headers)? In your explanation you jumped around between HIC and LIC and it was hard to follow.
Response: Thank you for your suggestion. We try our best to organize by sub-headings. However sometimes it was difficult to separate by HIC vs. LMIC due to formatting issues with endnote (and references got messed up).
Similarly, you also tend to speak about children/adolescents vs adults. Consider structuring it something like HIC HIC diff in ages LMIC LMIC differences in age?
Response: Thank you see response above.
Figure 1: font size is small and quite hard to read in “Full-text excluded with reasons.” Could condense reasons and increase font (ex. Not published in English (n=15))
Response: We agree with you. We increase the font size and condensed the reasons.
Line 208: no need to repeat that all articles were published in English, this was stated in the methods and is most appropriate there.
Response: Thank you. We removed that all articles were published in English.
Lines 235 and 254 are extra
Response: Thank you and we removed the extra.
Table 1: I find the bullets unnecessary and distracting, recommend removing them
Response: We removed the bullets.
Table 1: I don’t think there is a way around this, but I found the education column confusing since the terminology changes country to country and study to study.
Response: We removed the education column to avoid confusion on terminology changes country to country.
Table 1: ref 29 – what does Non-Education mean?
Response: See response above – we removed education column
Table 1: ref 68 - I think you can just state college not CEGEP
Response: see response above – we removed education column
Table 1: FFQ – be consistent stating number of items (sometimes you do and sometimes you don’t)
Response: Thank you for your concern. Since studies varied on the number of items, we removed the items number.
Table 1: need to be consistent in communication of info and formatting!
Example of the age column the format changes (what is this format saying 12-17: 15.1±1.9y?)
Response: We appreciate your concern. However, studies other report by age range or by mean (SD/SE). So, we opt to leave this as it was reported in the studies. If editor request, we are prepared to remove the column of age as well.
There are abbreviations that are not defined. In age, state M… what is that?
Sometimes say y and sometimes yo
consider footnotes stating categorical info is expressed … and continuous is expressed …. As well as footnotes with abbreviations.
Response: Thank you. We added the footnotes the abbreviations.
Line 263: This is the Food Pyramid from the US? If so, isn’t it fine to just say myPyramid?
Results: We appreciate your comment. However, other countries used the pyramid figure as a guideline, so we thought it would be better to differentiate it.
Table 2: Summary of results column is geared towards adherence, correct? Please provide a better heading, tailor these entries to be focused on adherence, and be consistent in the language in this column (instead of meeting recommendations, use adhering to recommendations)
Response: Thank you. We changed on the language.
Figure 2: consistent font style as article, have a * in title but not in the footnote, make sure the font size is readable, need to have titles on axis, and title of the figures should be more understandable (what does Global rating mean number of articles in range?… I don’t understand what this is referring to) and be able to stand alone.
Response: We apologized and provided the changes. We added a footnote (*) for what EEPHPP means. We removed the graphic that says the “global rating mean” as it was confusing for readers, and not all quality assessment tools for systematic reviews used this term.
Line 294: as mentioned in the methods, it is unclear as to how adherence was determined from all this data.
Response: We are thankful for your suggestion. We added a brief explanation of how adherence was measured in the methods section.
Discussion
Line 383: do not need this abbreviation, instead can just use the HIC and LMIC abbreviations. As stated previously, need to check your abbreviations.
Response: Thank you. This was also requested by the other reviewer.
Round 2
Reviewer 1 Report
Dear authors,
I am coming back with some additional comments on the revision #2.
Please check again the number of words in abstract. I counted 204.
Line 64-65: Please use the term across countries with different socio-economic status
Line 68-69: comparisson -> comparison. Refine the sentence: The comparison between HIC and LMIC adherence to national FBDGs will inform potential improvements in future dietary guidelines.
Line 103: FBDGs abbreviations was introduced in line 52. No need here again.
Line 106: 24h Recall -> 24h dietary recall (24hDR) use this abbreviation in line 399; spacing after full stop!
Line 109: Assed??? This sentence sound a bit like vicious circle. Please elaborate more precisely. Adherence to recommendations in national FBDGs was assessed on individual level or smt like that. Elaborate more on what was assessed, because there are many indicators and recommendations in FBDGs and they might not all be the same in every, so only some common points could be compared. Right?
Line 119: You have already introduced abbreviation for HIC and LMIC in introduction.
Line 212-223: Spacing between number and "years"; decimal places for number of populations in line 212-213, form of number in not consistent.
Line 357: women of childbearing age years -> women of child-bearing age
Line 367: Usually we observe intake of n-3 and n-6 and their ration, not n-9.
Line 374: Refine spacing in the brackets.
Line 416: Socio-demographic factors like what??? Please, emphasize which ones? There are many.
Author Response
Please check again the number of words in abstract. I counted 204.
Response: We appreciate your comment. I just checked the abstract count and for me, I have 193 words.
Line 64-65: Please use the term across countries with different socio-economic statuses.
Response: Thank you. We changed the term across countries with (…).
Line 68-69: comparisson -> comparison. Refine the sentence: The comparison between HIC and LMIC adherence to national FBDGs will inform potential improvements in future dietary guidelines.
Response: Thank you. We refine the sentence.
Line 103: FBDGs abbreviations was introduced in line 52. No need here again.
Response: We appreciate your comment. We remove the complete wording.
Line 106: 24h Recall -> 24h dietary recall (24hDR) use this abbreviation in line 399; spacing after full stop!
Response: Thank you. We provide the changes.
Line 109: Assed??? This sentence sound a bit like vicious circle. Please elaborate more precisely. Adherence to recommendations in national FBDGs was assessed on individual level or smt like that. Elaborate more on what was assessed, because there are many indicators and recommendations in FBDGs and they might not all be the same in every, so only some common points could be compared. Right?
Response: We appreciate your comment. We elaborate on this.
Line 119: You have already introduced abbreviation for HIC and LMIC in introduction.
Response: We remove full writing on HIC and LMIC.
Line 212-223: Spacing between number and "years"; decimal places for number of populations in line 212-213, form of number in not consistent.
Response: We provided the spacing and consistency for the decimals.
Line 357: women of childbearing age years -> women of child-bearing age
Response: Thank you. We provided the change.
Line 367: Usually we observe intake of n-3 and n-6 and their ration, not n-9.
Response: Thank you for the clarification. We agree with you. We made the correction.
Line 374: Refine spacing in the brackets.
Response: We appreciate your comment on this. However, is the standard of the reference used, and the endnote does not allow us to provide this change. We can provide this change when we convert a plain document. We afraid to convert to pain document and after this, we have to provide changes on references, and then it will be much harder for us to provide changes on references.
Line 416: Socio-demographic factors like what??? Please, emphasize which ones? There are many.
Response: Thank you. We provided an example of socio-demographic factors.
Reviewer 2 Report
Thank you for your revision, please find my comments below.
General: Please be consistent with whether or not you put a '.' in initials.
Line 65: please clarify what "these adherence" is.
Line 212 & 213: state that the sample size is people/individuals
Table 1 and 2: Why is the order of references listed as it is? It seems random and would therefore be helpful to have some order.
Author Response
General: Please be consistent with whether or not you put a '.' in initials.
Response: We are now consistent with the ‘ ‘. We removed all the ‘ ‘.
Line 65: please clarify what "these adherence" is.
Response: We are sorry for that. We provided clarification for “this adherence
“.
Line 212 & 213: state that the sample size is people/individuals.
Response: We stated that the sample size is individuals.
Table 1 and 2: Why is the order of references listed as it is? It seems random and would therefore be helpful to have some order.
Response: We appreciate your comment. On tables 1 and 2 we added references based on the date (year) of publication.